# FEATURE PARTITIONING FOR EFFICIENT MULTI-TASK ARCHITECTURES

## ABSTRACT

Multi-task learning promises to use less data, parameters, and time than training separate single-task models. But realizing these benefits in practice is challenging. In particular, it is difficult to define a suitable architecture that has enough capacity to support many tasks while not requiring excessive compute for each individual task. There are difficult trade-offs when deciding how to allocate parameters and layers across a large set of tasks. To address this, we propose a method for automatically searching over multi-task architectures that accounts for resource constraints. We define a parameterization of feature sharing strategies for effective coverage and sampling of architectures. We also present a method for quick evaluation of such architectures with feature distillation. Together these contributions allow us to quickly optimize for parameter-efficient multi-task models. We benchmark on Visual Decathlon, demonstrating that we can automatically search for and identify architectures that effectively make trade-offs between task resource requirements while maintaining a high level of final performance.

## 1 INTRODUCTION

Multi-task learning allows models to leverage similarities across tasks and avoid overfitting to the particular features of any one task (Caruana, 1997; Zamir et al., 2018). This can result in better generalization and more robust feature representations. While this makes multi-task learning appealing for its potential performance improvements, there are also benefits in terms of resource efficiency. Training a multi-task model should require less data, fewer training iterations, and fewer total parameters than training an equivalent set of task-specific models. In this work we investigate how to automatically search over high performing multi-task architectures while taking such resource constraints into account.

Finding architectures that offer the best accuracy possible given particular resource constraints is nontrivial. There are subtle trade-offs in performance when increasing or reducing use of parameters and operations. Furthermore, with multiple tasks, one must take into account the impact of shared operations. There is a large space of options for tweaking such architectures, in fact so large that it is difficult to tune an optimal configuration manually. Neural architecture search (NAS) allows researchers to automatically search for models that offer the best performance trade-offs relative to some metric of efficiency.

Here we define a multi-task architecture as a single network that supports separate outputs for multiple tasks. These outputs are produced by unique execution paths through the model. In a neural network, such a path is made up of a subset of the total nodes and operations in the model. This subset may or may not overlap with those of other tasks. During inference, unused parts of the network can be ignored by either pruning out nodes or zeroing out their activations (Figure 1). Such architectures mean improved parameter efficiency because redundant operations and features can be consolidated and shared across a set of tasks.

We seek to optimize for the computational efficiency of multi-task architectures by finding models that perform as well as possible while reducing average node use per task. Different tasks will require different capacities to do well, so reducing average use requires effectively identifying which tasks will ask more of the model and which tasks can perform well with less. In addition, performance is affected by how nodes are shared across tasks. It is unclear when allocating resources whether sets of tasks would benefit from sharing parameters or would instead interfere.

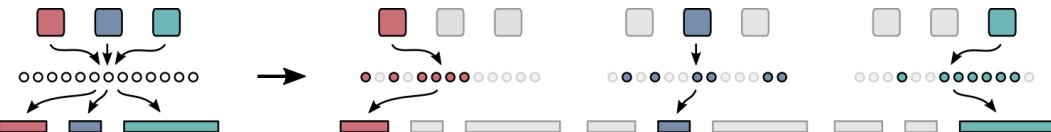

Figure 1: Feature partitioning can be used to control how much network capacity is used by tasks, and how much sharing is done across tasks. In this work we identify effective partitioning strategies to maximize performance while reducing average computation per task.

When searching over architectures, differences in resource use can be compared at different levels of granularity. Most existing work in NAS and multi-task learning searches over the allocation and use of entire layers (Zoph & Le, 2016; Fernando et al., 2017; Rosenbaum et al., 2017), we instead partition out individual feature channels within a layer. This offers a greater degree of control over both the computation required by each task and the sharing that takes place between tasks.

The main obstacle to address in searching for effective multi-task architectures is the vast number of possibilities for performing feature partitioning as well as the significant amount of computation required to evaluate and compare arrangements. A naive brute search over different partitioning strategies is prohibitively expensive. We leverage our knowledge of the search space to explore it more effectively. We propose a parameterization of partitioning strategies to reduce the size of the search space by eliminating unnecessary redundancies and more compactly expressing the key features that distinguish different architectures.

In addition, the main source of overhead in NAS is evaluation of sampled architectures. It is common to define a surrogate operation that can be used in place of training a full model to convergence. Often a smaller model will be trained for a much shorter number of iterations with the hope that the differences in accuracy that emerge early on correlate with the final performance of the full model. We propose a strategy for evaluating multi-task architectures using feature distillation which provides much faster feedback on the effectiveness of a proposed partitioning strategy while correlating well with final validation accuracy.

In this work we provide:

- a parameterization that aids automatic architecture search by providing a direct and compact representation of the space of sharing strategies in multi-task architectures.

- an efficient method for evaluating proposed parameterizations using feature distillation to further accelerate the search process.

- results on Visual Decathlon (Rebuffi et al., 2017) to demonstrate that our search strategy allows us to effectively identify trade-offs between parameter use and performance on diverse and challenging image classification datasets.

## 2  RELATED WORK

**Multi-Task Learning:** There is a wide body of work on multi-task learning spanning vision, language, and reinforcement learning. The following discussion will center around designing multi-task architectures for deep learning in vision (Ruder, 2017; Caruana, 1997; Zhang & Yang, 2017). There are many obstacles to overcome in multi-task architecture design, but the most pressing concerns depend largely on how the problem setting has been defined. Two distinguishing factors include:

- *Task Ordering:* Are all tasks available at all times or are they presented one after the other?

- *Fixed vs Learned Strategies*: Is a uniform strategy applied across tasks or is a task-specific solution learned?

The former is important as work in which tasks are presented sequentially must address catastrophic forgetting (French, 1999). This is less of a concern in our work as we train on all tasks at once. As for the latter, finding a solution tuned to a specific set of tasks requires the same sort of outer-loop optimization seen in neural architecture search (Zoph & Le, 2016) which is time-consuming and expensive. The contributions presented in this work seek to make this process more manageable.

**Multi-Task Architectures:** A strong baseline for multi-task architectures is the use of a single shared network (Caruana, 1997; Kaiser et al., 2017). Deep networks are overparameterized in such a way that the same layers can be applied across different domains while producing features that are useful to different ends. Using a shared architecture is common in reinforcement learning to train a single agent to perform many tasks with a uniform observation and action space (Espeholt et al., 2018; Sharma et al., 2017). A common technique to train single shared models well in both reinforcement learning and vision is distillation of multiple models into one (Beaulieu et al., 2018; Yim et al., 2017; Rusu et al., 2015; He et al., 2018; Chou et al., 2018).

In work where tasks are presented sequentially, the focus is often to build on top of an existing network while not disrupting its ability to solve its original task (Mallya & Lazebnik, 2018; Rebuffi et al., 2018; Rosenfeld & Tsotsos, 2017). Currently, many methods freeze the network's weights so they are not changed while learning a new task. Examples include masking out specific filter weights (Mallya & Lazebnik, 2018) or introducing auxiliary layers (Rebuffi et al., 2018). Another approach is to dynamically expand a network with additional capacity for each new task (Yoon et al., 2018). All of these methods build on top of a fixed model, meaning that new tasks must perform the computation required for the original task as well as take additional steps to be task-specific.

It is also common to build multi-task architectures from sets of layers that are run in parallel (Misra et al., 2016; Rosenbaum et al., 2017; Fernando et al., 2017; Meyerson & Miikkulainen, 2017). Cross-stitch networks compute activations for each task as a learned weighted sum across these layers (Misra et al., 2016). This sort of soft attention over features can be seen in other multi-task architecture work as well (Liu et al., 2018; Ruder et al., 2017). There are approaches to search over paths through these layers such that each task has a unique, optimal execution path (Rosenbaum et al., 2017; Fernando et al., 2017). Similar to work in single-task NAS, the best path is found by either reinforcement learning or evolutionary algorithms (Fernando et al., 2017; Liang et al., 2018). The optimal trade-offs in parameter sharing may occur at a more fine-grained level than entire layers, so instead of working with parallel blocks of layers we divide up individual feature channels.

**Neural Architecture Search:** There are three main areas in which contributions are made for more effective architecture search: search space, optimization, and sample evaluation.

*Search space:* With a well-designed search space, it is possible to randomly sample and arrive at high performing solutions (Li & Talwalkar, 2019; Liu et al., 2017b). In general, NAS operates in a discrete space where entire layers are included or not. We instead propose a continuous search space where slight changes can be made in how resources are allocated across tasks. This allows alternatives for optimization that would not apply in other NAS work.

*Optimization:* Leading approaches either use reinforcement learning or genetic algorithms for NAS (Zoph & Le, 2016; Real et al., 2018; Pham et al., 2018). This search is difficult and the trade-offs between approaches are unclear (Li & Talwalkar, 2019). We test the effectiveness of random sampling and evolutionary strategies optimization (Mania et al., 2018; Wierstra et al., 2008).

*Evaluating Samples:* Training a model to convergence is time-consuming and resource intensive. It is not realistic to sample thousands of architectures and train them all. Instead one must use a cheaper form of evaluation. Some options include preserving weights across samples for faster training (Pham et al., 2018), successive halving (Kumar et al., 2018), progressive steps to increase complexity (Liu et al., 2017a), as well as techniques to model the expected performance of sampled architectures (Deng et al., 2017; Brock et al., 2017; Baker et al., 2017). It is unclear how well surrogate functions correlate with final model performance (Zela et al., 2018). We investigate the use of distillation for performing this evaluation.

## 3 MULTI-TASK FEATURE PARTITIONING

Sharing in the context of multi-task architecture search is often adjusted at the level of individual layers (Rosenbaum et al., 2017; Fernando et al., 2017), but given that state-of-the-art architectures work so well across datasets and tasks we choose to preserve the layer level execution path for each task and focus on sub-architectural changes that can be made. In this case, by deciding whether or not individual feature channels within each layer are available. This way, every task experiences the exact same organization of layers, but a unique calculation of intermediate layer features.

Concretely, given a feature tensor $\mathsf{F} \in \mathbb{R}^{c \times h \times w}$ we define a binary mask $\boldsymbol{m}_f \in \{0,1\}^c$ for each task, and during a forward pass of the network multiply $\mathsf{F}$ by $\boldsymbol{m}_f$ to zero out all channels not associated with a particular task. We further define a mask for the backward pass, $\boldsymbol{m}_b \in \{0,1\}^c$, whose non-zero elements are a subset of the non-zero elements in $\boldsymbol{m}_f$. Gradients are calculated as usual through standard backpropagation, but any weights we wish to leave unchanged will have their gradients zeroed out according to $\boldsymbol{m}_b$.

Together, these masks can capture the training dynamics seen in many existing multi-task architecture designs (Rosenbaum et al., 2017; Rebuffi et al., 2018). For example, one can devote an outsized proportion of features to a task like ImageNet classification, then make these features available during forward inference on a new smaller dataset. A backward mask, $\boldsymbol{m}_b$, can then be defined to ensure that the ImageNet weights remain untouched when finetuning on the new task.

There are a number of advantages to allocating resources at the channel level. There is enough flexibility to allow fine-grained control allotting specific weights to particular subsets of tasks. And after training, it is straightforward to prune the network according to any mask configuration. Meaning for simple tasks that only require a small subset of channels we can use a fraction of compute at test time while still leveraging the advantages of joint training with other tasks.

An important implementation detail is that masks are applied every other layer. Consider, for example, making a task use half the model. You might think to set half the values of $\boldsymbol{m}_f$ to one and apply it to each layer. But that would mean $\frac{c}{2}$ inputs and $\frac{c}{2}$ outputs at each layer which only uses one quarter of the original model. Instead, applying a mask at every other layer produces the desired behavior of allocating half the model to the task.

### 3.1 PARTITIONING PARAMETERIZATION

Now that we have decided to partition up feature channels, how do we go about finding the best masks for each task? Consider defining a binary matrix that specifies all partitioning masks: $\boldsymbol{M} \in \{0,1\}^{c \times n}$, where $c$ is the number of feature channels and $n$ is the total number of tasks. A direct search over this matrix is problematic. It is not straightforward to optimize over a space of many discrete values, and one must account for significant redundancy given that all permutations of channels are equivalent. Moreover, naive random sampling would never cover the full space of partitioning strategies (consider the probability of randomly sampling two masks $\boldsymbol{m}$ that were mutually exclusive). In order to see diverse degrees of feature sharing, the overlap of channels between masks must be explicitly accounted for.

Thus instead of searching over $\boldsymbol{M}$, we propose searching directly over the features of $\boldsymbol{M}$ that determine performance: 1) the number of feature channels used by each task, and 2) the amount of sharing between each pair of tasks. The former decides the overall capacity available for the task while the latter shapes how tasks help or interfere with each other. We explicitly parameterize these factors in a matrix $\boldsymbol{P}$. We can compute $\boldsymbol{P} = \frac{1}{c}\boldsymbol{M}^T\boldsymbol{M}$, where the diagonal elements of $\boldsymbol{P}$ provide the percentage of feature channels used by each task and the off-diagonal elements define the percentage of overlapping features between task pairs.

When sampling new partitioning strategies, we sample directly from $\boldsymbol{P}$ and identify a corresponding mask $\boldsymbol{M}$ to match it. To remove some ill posed parts of the space we take additional steps to adjust $\boldsymbol{P}$. More details of this process as well as how we derive $\boldsymbol{M}$ from $\boldsymbol{P}$ can be found in the appendix.

This representation has a number of distinct advantages. It is not tied to the number of channels in a given layer, so a single parameterization can be used for layers of different sizes. It is low dimensional, particularly since $n$ is typically much smaller than $c$. And it is interpretable, providing a clear impression of which tasks require more or less network capacity and which tasks train well together. Moreover we get an immediate and direct measurement of average node usage per task, this is simply the mean of the diagonal of $\boldsymbol{P}$. We will use this metric to compare the resource efficiency of different proposed partitioning strategies.

## 4 OPTIMIZATION STRATEGY

In order to optimize over different parameterizations $\boldsymbol{P}$, there are two key ideas to cover: how we choose samples from the space and how we evaluate and compare different samples.

## 4.1 SEARCH STRATEGIES

We treat our search setting as a black box optimization problem where given a particular parameterization we have a function which returns a score assessing its quality. Based on this score we can then choose how to further sample new parameterizations. We investigate two strategies for finding good constraint matrices.

*Random sampling:* The first is to simply randomly sample values. This has already been demonstrated to serve as a strong baseline in some architecture search work (Li & Talwalkar, 2019; Liu et al., 2017a). With the low dimensionality of the matrix as well the additional steps taken to preprocess constraints, it is not unreasonable that much of the space can be covered with random samples. Random samples serve well to map out large swaths of the search space and identify the principle choices that affect final performance. Concretely, a random matrix $P$ can be sampled with values taken uniformly from 0 to 1. If a particular resource target is desired, it is trivial to bias or restrict samples to a specific range of parameter usage.

*Evolutionary strategies:* Because $P$ is continuous, it is possible to search over parameterizations with gradient-based optimization. We run experiments using evolutionary strategies [1]. More specifically, we use a simple implementation with the modifications as described by Mania et al. (2018). A gradient direction is approximated by sampling several random directions in the parameter space and computing finite differences to see which directions seem most promising. A weighted average is then computed across all directions to calculate the gradient to update the current parameters.

A key feature of our approach is that we modify the algorithm to prioritize parameterizations that use as few channels as necessary per task. An additional L2 weight regularization term is added to the parameters on the diagonal of $P$. This serves to reduce the number of channels used by each task, in particular those that can be pulled down without affecting the overall accuracy and performance of the model. By controlling the strength of this regularization we can tune the importance of resource efficiency in the search process.

Using this optimization strategy is only possible because of the parameterization defined in 3.1. Approximating gradients make sense in the context of the continuous constraints defined in $P$, and we can more effectively explore the space of multi-task architectures using this signal. This is different from existing architecture search work where search decisions correspond to the coarse selection of entire computational blocks and their connections to each other.

## 4.2 SAMPLE EVALUATION

Finally, we must evaluate different partitioning schemes. But as discussed, determining the relative effectiveness of one partitioning over another by training models to convergence is expensive. One possible strategy is to train models for a short period of time assuming that the relative differences in performance that appear early in training should correlate well with differences in performance when trained for longer. We instead propose to use feature distillation to observe the representational capacity of a partitioned layer. We test how well shared multi-task layers can reproduce the activations of corresponding single-task layers. By only focusing on a few layers, we reduce total computation and the number of weights that need to be tuned. In addition, directly distilling to intermediate layer activations provides a more direct training signal than a final classification loss.

Given a proposed partitioning mask, we initialize new layers to be distilled and load reference models for each target task. Input to the layer is generated by passing through the task-specific pretrained model up to a target depth. The resulting features are then passed through the subsequent layers of the pretrained model as well as the new shared layers. In the new layers, intermediate features are masked according to the proposed partitioning. This procedure is illustrated in Figure 2.

We use a mean-squared error loss to supervise the shared layers such that their output features match those produced by the reference teacher models. We can measure the effectiveness of this distillation by replacing the original pretrained layers with the new shared layers and measuring the updated model accuracy.

---

[1]This can also be referred to as "random search" (Mania et al., 2018), but we will instead use "evolutionary strategies" to avoid confusion with random sampling which is also called random search in existing NAS work (Li & Talwalkar, 2019).

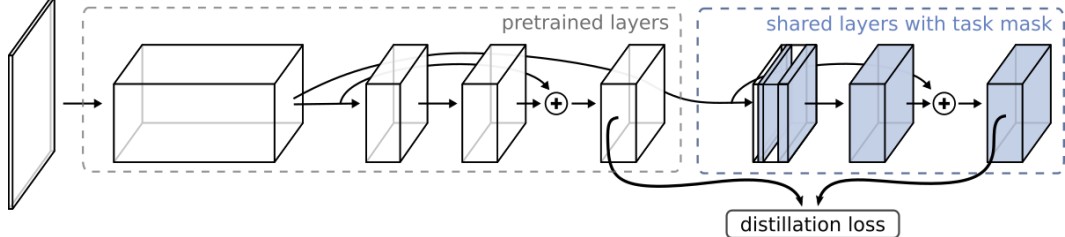

Figure 2: Multi-task distillation: At a given step, a teacher network pretrained on a single task is chosen from the available tasks. Features at a target depth are produced then passed through the next residual block of both the teacher and student (where a mask is applied). Features are then compared using a standard MSE loss. This process can be applied to multiple layers at once.

It is important to emphasize that we are not using distillation to get a final multi-task model as we do not want to be limited by the performance of individual pre-trained models. Instead, distillation serves as a proxy task for quickly evaluating partitioning strategies. We do not run the distillation process to convergence, only for a brief interval, and it serves as a sufficient signal to provide feedback on different parameterizations. This leads to a dramatic reduction in the time required to evaluate a particular masking strategy.

## 5 EXPERIMENTS

We run a number of experiments to investigate the role our proposed parameterization and distillation play in finding multi-task architectures that minimize task computation and parameter use while achieving high accuracy. All experiments are performed using the Visual Decathlon dataset (Rebuffi et al., 2017). Visual Decathlon is composed of many well-established computer vision classification datasets of various sizes and respective difficulties. There is sufficient diversity that it is difficult to determine which datasets would benefit from more or less network capacity and parameter sharing.

We investigate how a model performs when trained on nine Decathlon tasks at once (all datasets except for ImageNet). We initialize a shared ResNet model with a separate fully connected layer output for each task. To simplify experiments, we freeze the first two-thirds of the model and only apply feature partaining to the last third. For training, we alternate mini-batches sampled from each dataset and apply a standard cross-entropy classification loss at the appropriate task specific output. More thorough implementation and experiment details can be found in the appendix.

It is unclear that performance will necessarily be better with any feature restriction as opposed to using the full model for all tasks. One question is whether partitioning well leads to a reduction in interference across tasks and perhaps improved performance. In addition, we wish to see the overall relationship between performance and feature restriction in this multi-task setting. What's the best performance possible as average feature use is reduced further and further?

### 5.1 DISTILLATION

Before performing our search we need to know that given a sampled set of feature masks $M$, distillation performance correlates well with the final accuracy of a model trained to convergence. This will determine whether our proposed distillation is a reasonable surrogate in lieu of full training. The higher the correlation between the two, the more confidence we can place in our search process.

When performing distillation we initialize the child layers with pretrained layers from an ImageNet model, since the parent single-task networks have also been initialized from the same model. This accelerates the distillation process. The whole process takes just one minute on a P100 GPU. Further details are available in the appendix.

We sample many random partitioning masks and run both the distillation procedure and full training to convergence. As a baseline, we also see how well final validation accuracy compares to accuracies seen earlier in training. We compare to the accuracies after 5k and 10k iterations (corresponding to 5 and 10 minutes of training). As seen in Table 1, the distillation procedure (which takes a fraction

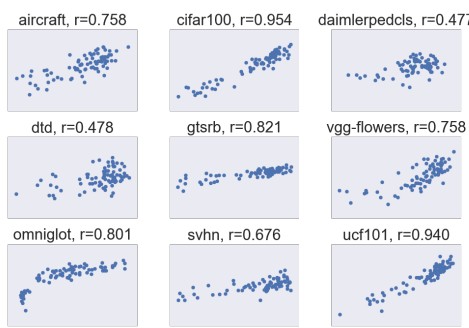

| Dataset | 5k iters | 10k iters | dist |
|---|---|---|---|
| aircraft | 0.658 | 0.690 | **0.758** |
| cifar100 | 0.928 | 0.927 | **0.954** |
| daimlerpedcls | 0.018 | 0.160 | **0.477** |
| dtd | 0.368 | 0.436 | **0.478** |
| gtsrb | 0.714 | 0.687 | **0.821** |
| vgg-flowers | 0.703 | 0.692 | **0.758** |
| omniglot | 0.871 | **0.906** | 0.801 |
| svhn | 0.485 | **0.761** | 0.676 |
| ucf101 | 0.811 | 0.732 | **0.940** |

Figure 3: Correlation between distillation accuracy and final validation accuracy.

Table 1: Correlation to final validation accuracy (distillation vs training 5k and 10k iterations).

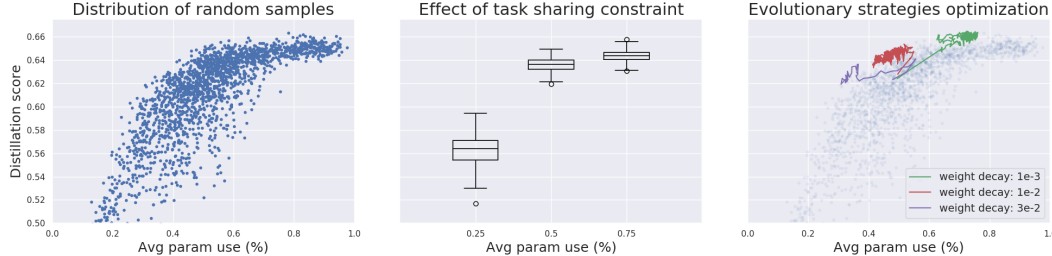

Figure 4: (left) Distribution of performance of random partitioning strategies; (middle) Distribution of performance when fixing channel use but varying sharing across tasks; (right) Optimization trajectories of ES with different degrees of regularization

of the time) correlates higher with final accuracy. This allows us to sample and compare many more parameterizations during our search process and have more confidence that the top performing parameterizations will do well when training a full model.

## 5.2 ARCHITECTURE SEARCH

**Randomly sampling parameterizations:** To map out the performance of different partitioning strategies we sample random parameterizations and plot distillation performance against the average percentage of allocated features (the mean of the diagonal of $P$) in Figure 4 (left). From the distribution of random samples we get an impression of the best performance possible at different degrees of resource use. The combination of fast feedback with distillation plus effective search space coverage with our proposed parameterization produces this information with less samples and in less time. At high levels of average feature use, choice of partitioning can only make so much of a difference. We are interested in the opposite - how well we can do when restricting task computation as much as possible. Here, partitioning well is necessary to achieve high performance.

Also it is important to note that explicitly specifying the degree of sharing between tasks is critical. This is shown in the middle plot of Figure 4. We evaluate different partitioning strategies with three fixed sets of values for the diagonal of $P$, only adjusting the amount of sharing that takes place between tasks. There can be a significant difference in performance in each of these cases, and as expected, sharing affects performance more and more as average parameter use goes down as there is more flexibility when choosing how to overlap features. It is important that feature sharing is parameterized when doing any sort of optimization.

Finally, we look at per-task results (Figure 5). In this particular setting, every task benefits from using as many features as possible. This may have to do with using a model pretrained on ImageNet, but it makes sense that tasks benefit from using as many features as they can. An important facet to this is how much of those features are shared. As average parameter usage increases across tasks (indicated by a lighter color), individual tasks suffer as they now share more of their features and must deal with the interference of other tasks.

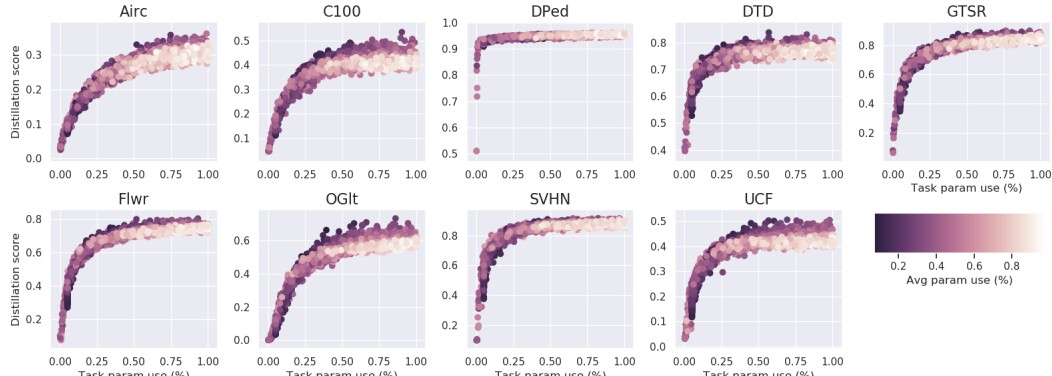

Figure 5: Distillation performance per task as a function of parameters allocated to that task. Color indicates the average parameters used across all tasks, notice as more parameters are used overall (lighter color) individual task performance drops.

| Partitioning | Airc. | C100 | DPed | DTD | GTSR | Flwr | OGlt | SVHN | UCF | Total |
|---|---|---|---|---|---|---|---|---|---|---|
| independent | 53.2 ± 0.2 | 74.1 ± 0.2 | 97.8 ± 0.3 | 49.4 ± 0.3 | 99.7 ± 0.0 | 78.3 ± 0.2 | 87.7 ± 0.1 | 94.3 ± 0.1 | 77.1 ± 0.3 | 79.1 ± 0.1 |
| share half | 53.1 ± 0.3 | 75.4 ± 0.2 | 98.2 ± 0.1 | 49.5 ± 0.3 | 99.8 ± 0.0 | 78.6 ± 0.3 | 87.7 ± 0.1 | 94.0 ± 0.1 | 78.2 ± 0.2 | 79.4 ± 0.1 |
| share all | 54.4 ± 0.2 | 76.3 ± 0.3 | 98.1 ± 0.1 | 50.0 ± 0.3 | 99.7 ± 0.0 | 78.9 ± 0.6 | 87.7 ± 0.1 | 94.1 ± 0.1 | 79.0 ± 0.3 | 79.8 ± 0.2 |
| es (wd 1e-3) | 54.2 ± 0.2 | 77.0 ± 0.1 | 97.6 ± 0.2 | 49.7 ± 0.1 | 99.8 ± 0.0 | 81.3 ± 0.4 | 88.3 ± 0.0 | 94.0 ± 0.1 | 79.4 ± 0.1 | 80.1 ± 0.1 |

Table 2: Validation results on Visual Decathlon.

**Evolutionary strategies:** As mentioned above, the distribution of random samples gives an immediate impression of the best level of performance possible as a function of average parameter use. Evolutionary strategies provides a means to more directly push this edge of performance even further. We visualize the search process by plotting samples over the course of optimization overlaid over the distribution of samples found by random sampling (Figure 4 (right)). ES optimization quickly identifies samples that provide the best accuracy given their current level of parameter use and densely samples in this space, making slight changes for any last available improvements to performance. Furthermore, adjusting the weight decay penalty used during optimization controls the resource use of the final partitioning strategy. This allows us to easily tune the optimization to reach the best architecture that meets specific resource needs.

The best parameterization found with evolutionary strategies outperforms a number of baselines for performing partitioning as seen in Table 2. We compare across several strategies with different degrees of feature use and sharing. We measure validation accuracy of models trained to convergence (averaged over five trials). These baselines include: independent partitions that split features evenly across tasks, sharing half of available feature channels and splitting the rest, and finally, sharing all feature channels. In line with our random sampling, the more channels given across tasks, the better performance. Sharing everything does the best amongst these baselines. However, using the parameterization found from our optimization both reduces average channel use and achieves better performance overall.

We see that there exist partitioning strategies that cut down average feature dramatically while still maintaining the same overall performance. This is in large part due to simple tasks that only need a small fraction of channels (DPed for example in Fig 5). By taking away the interference caused by these simpler tasks, harder tasks stand to gain more and that can be seen in Table 2 with tasks like CIFAR100, Flowers, and Omniglot seeing the largest gains from an effective partitioning strategy.

## 6  CONCLUSION

In this work we investigate efficient multi-task architecture search to quickly find models that achieve high performance under a limited per-task budget. We propose a novel strategy for searching over feature partitioning that automatically determines how much network capacity should be used by each task and how many parameters should be shared between tasks. We design a compact representation to serve as a search space, and show that we can quickly estimate the performance of different partitioning schemes by using feature distillation.

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

# A APPENDIX

## A.1 PARTITIONING PARAMETERIZATION

**Refining $P$:** We define a simple set of operations that convert from the raw search space $P$ to a constraint matrix $\widetilde{P}$ that is more likely to correspond to feasible masks. Given the knowledge that pairwise values of $P$ are conditioned on its diagonal terms, it is not possible for there to be more overlap between two tasks than the channels used by any one task. That is, no off-diagonal element $M_{ij}$ should be greater than the corresponding diagonal elements $M_{ii}$ and $M_{jj}$.

We remap all off-diagonal elements to appropriate values determined by the diagonal of the matrix. This means that for any off-diagonal element in $P$, 0 now maps to the minimum possible overlap and 1 to the maximum possible overlap of the two tasks. The procedure is defined as follows:

$$D = \text{diag}(P)\mathbf{1}^T \in \mathbb{R}^{n \times n} \tag{1}$$

$$P_{min} = \max(0, D + D^T - J) \tag{2}$$

$$P_{max} = \min(D, D^T) \tag{3}$$

$$\widetilde{P} = P \odot I + (P \odot (P_{max} - P_{min}) + P_{min}) \odot (J - I) \tag{4}$$

where $\mathbf{1}$ and $J$ are the column vector and the matrix of ones respectively, and $\odot$ represents Hadamard product.

**Deriving $M$ from $\widetilde{P}$:** Now, we must find a feasible mask $M$ to satisfy the constraints specified in $\widetilde{P}$. This can be formulated as a mixed integer programming problem. To show that, let $\widetilde{P} \in [0, 1]^{n \times n}$ denote a given parameterization of constraints, we are interested in determining a binary mask $M$ by minimizing:

$$\text{minimize} \sum_{i=1}^{n} \phi_i + \sum_{i=1}^{n}\sum_{j=1}^{n} \xi_{ij} \qquad\qquad \text{subject to:} \tag{5}$$

$$\sum_{k=1}^{c} M_{ki} \leq \widetilde{P}_{ii} + \phi_i \qquad\qquad \forall i \in [1, n] \tag{6}$$

$$\sum_{k=1}^{c} t_{ijk} \leq \widetilde{P}_{ij} + \xi_{ij} \qquad\qquad \forall i, j \in [1, n] \tag{7}$$

$$t_{ijk} \leq \frac{1}{2}(M_{ki} + M_{kj}) \qquad\qquad \forall i, j \in [1, n] \tag{8}$$

$$t_{ijk} \geq M_{ki} + M_{kj} - 1 \qquad\qquad \forall i, j \in [1, n] \tag{9}$$

$$t_{ijk}, M_{ki} \in \{0, 1\}, \phi_i, \xi_{ij} \geq 0 \qquad\qquad \forall k \in [1, c] \tag{10}$$

We employ two techniques. First we introduce two slack variables $\phi$ and $\xi$ to relax the unary task constraint in equation 6 and pairwise task constraints in equation 7. Second, we convert the nonlinear constraints between tasks by introducing the binary auxiliary variable $t$. For each pair of tasks $(i, j)$, in a perfect solution without slack variables, we have:

$$M^T M = c\widetilde{P} = \sum_{i=1}^{n}\sum_{j=1}^{n}\sum_{k=1}^{c} t_{ijk} \tag{11}$$

Two tricks are used: auxiliary variable $t$ for non-linear constraint and slack variables. The search space now is significantly reduced, and this becomes a mixed integer programming problem which can be conveniently solved by off-the-shelf solvers.

Figure 6: Measuring correlation of feature activations across three blocks of task-specific ResNet models given a shared input image. Even though no restriction is made when finetuning individual models, the features produced are highly correlated through the first two-thirds of the model, and greater task differentiation is not seen until the end. Differentiation in early blocks would normally occur due to different batch normalization statistics across datasets, but that is controlled for here.

## A.2 Additional experiment details

**Multi-task training:** For full model training, we use a batchsize of 64 with SGD and momentum at a learning rate of 0.05 for 100k iterations, dropping to a learning rate of 0.005 at iteration 75k. All training was done on a single Nvidia P100 GPU. We followed the exact training, validation, and test splits provided by Visual Decathlon.

Several steps are taken to ensure that a model trained simultaneously on multiple tasks converges well:

- *Batch normalization:* We maintain separate batch normalization statistics per task as done in (Rebuffi et al., 2017). This adds minimal parameter overhead and accelerates training.

- *Momentum:* We maintain separate gradient statistics when using momentum with SGD. This is important given our use of feature partitioning. At any given training step, we do not want the unused weights associated with other tasks to be updated.

- *Training curriculum:* Rather than uniformly sampling across tasks we apply a simple strategy to choose mini-batches from tasks inversely proportional to their current training accuracy. Tasks that lag behind get sampled more often. Evidence for this approach has been demonstrated in a multi-task reinforcement learning setting (Sharma et al., 2017). We find a curriculum over tasks more effective than a curriculum over individual samples (Jiang et al., 2018).

- *Pretrained ImageNet model:* All experiments are performed with a pretrained ImageNet model. The model is trained using the PyTorch implementation made available by Rebuffi et al. (2018). Because ImageNet is orders of magnitude larger than any of the other datasets in Decathlon and takes much longer to train we exclude it in our partitioning experiments to focus on the interactions of other datasets.

The main hyperparameters that determine performance were learning rate and a temperature term that controlled the task sampling curriculum. This temperature term determines whether mini-batches are sampled uniformly across tasks or whether tasks with low training accuracy are weighted more heavily. For both hyperparameters we arrive at the final value by a simple grid search.

Final validation accuracy reported in Table 2 (in the main paper) is averaged across 5 trials.

**Frozen layers:** To further simplify experiments, we freeze and share the first two-thirds of the network. Partitioning is thus only performed on the last third of the model. By only updating the weights of the last block, we focus attention on the layers where task-specific features are most important without restricting the model's representational capacity to fit each task.

In all of our experiments we use an ImageNet-pretrained ResNet model made up of three computational blocks with four layers each. We freeze the first two computational blocks and only perform partitioning on the last set of layers. The justification for this stems from analysis performed with individual single task models. We compare feature differences across finetuned task-specific models. These models were trained with no restrictions initialized from an ImageNet model until converging to high accuracy on some target task. Because we start with a pretrained model we can make mean-

ingful comparisons of each model's channel activations to see how task feature use diverges after finetuning.

We compare intermediate task features after passing in a shared image into every model. An important detail here is that we control for the batch normalization statistics associated with the dataset that the image is sampled from. The subsequent features produced by each model are almost identical all the way up through the first two-thirds of the model. Aside from subtle differences, task-specific differentiation did not occur until the final third of the model where features were still somewhat correlated but differed dramatically model to model. This is visualized in Figure 6. Because of this we decided the task-specific differentiation afforded by feature partitioning would not be as important in earlier stages of the model, and experiments would be more informative and also faster to run while focusing only on the last set of layers.

**Distillation details:** We do not use the accuracy-based curriculum used in normal training during distillation and instead alternate mini-batches uniformly across each task. Distillation training is done for a brief 3000 iterations with a batch size of 4 and a learning rate of 1 which is dropped by a factor of 10 at iteration 2000.

Distillation is done on the last four ResNet layers at once to match the final training setting as closely as possible. All scores reported when performing distillation are averaged across three trials.

**Evolutionary strategies details:** The optimization curves shown in the paper are from runs that have each taken 1000 samples, these were performed on machines with 4 P100 GPUs. Given that sample evaluation takes roughly a minute, the whole procedure takes just over four hours.

At each step, 16 random parameter directions are sampled and these are both added and subtracted from the current parameterization $P$ to produce 32 new samples to evaluate. A gradient is calculated based on the results of these samples, and a gradient descent step is applied to the current parameters with a learning rate of 0.1. Both clipping and a sigmoid operation were tested to ensure that values remain between 0 and 1 with no discernible difference in optimization performance.

