# OpenReview forum: "Feature Partitioning for Efficient Multi-Task Architectures"
_ICLR.cc/2020/Conference — Reject_

### Official Review · AnonReviewer2 · 2019-10-21
**Official Blind Review #2**

**Rating:** 6

**Review:**

This paper studies multi-task learning (MTL) from the deep learning perspective where a number of layers are shared between tasks followed by specific heads for each task. One of the main challenges in this problem is to decide the best configuration among a large number of possible ones (e.g., the number of layers , number of neurons, when to stop the shared part of the network). In this paper, the authors fix the network architecture, and learn which filters (among the already learned ones) should be dedicated to (and hence fine-tuned for) a specific, and which ones should be shared between multiple tasks.

Instead of deciding on other hyper-parameters such as the number of layers, the authors chose to study how to efficiently share the capacity of the network: to decide which filters should be used for which tasks, and which filters should be shared between tasks.
Specifically, this is controlled by task specific binary vectors which get multiplied with feature activations for each task, hence blocking or allowing the signal to pass for a specific filter. In addition, they define a different set of binary vectors for the foreground and background passes. This allows simpler tasks to benefit from features learnt from more complicated tasks such as ImageNet classification while avoiding ‘catastrophic forgetting’ at the same time.

Moreover, the authors develop a simple yet elegant strategy to reduce their parameter search space (by using the matrix P which controls the percentage of filters used per task + the percentage of filters shared between each pair of tasks) and quickly evaluate the performance of each configuration (using distillation). The advantages of these approaches are well discussed and validated quantitatively.

The paper is well written and the approach itself appears to be sound and it led to improvement over independent task estimator.  However, I am mostly concerned about the experimental setting: there are no comparisons with any other MTL algorithm.

The authors perform a search over the matrix P, which is similar to neural architecture search over the entire possible ways of sharing the capacity of a network. This could potentially lead to improvement beyond multi-task learning. Experimental comparison on this could be provided.
I think the paper will make a strong case if it is compared with existing deep MTL algorithms including [Misra et al: Cross-stitch networks for multi-task learning]. In addition, the network seems to share a similar spirit with [Mallya et al: PackNet: Adding Multiple Tasks to a Single Network by Iterative Pruning], in that they also share the capacity of the network between tasks, and hence a comparison here seems reasonable.

Overall, I think this paper makes a borderline case.

Other comments:
In the supplementary material, providing a detailed description of the algorithm (e.g., pseudo code and an accompanying discussion) that calculates the matrices M from P could help reproduce and build upon the experiments reported in the paper. I wonder if M is uniquely defined from M.

**Experience Assessment:**

I have published one or two papers in this area.

**Review Assessment: Checking Correctness Of Derivations And Theory:**

I assessed the sensibility of the derivations and theory.

**Review Assessment: Checking Correctness Of Experiments:**

I assessed the sensibility of the experiments.

**Review Assessment: Thoroughness In Paper Reading:**

I read the paper at least twice and used my best judgement in assessing the paper.

---

> ### Author Response · Authors · 2019-11-15
> **reply**
>
> Thank you for all of your comments and taking the time to review.
>
> Cross-stitch networks and PackNet would make for interesting comparisons. While we do not have experiments to compare to these methods, it is worth noting some differences and advantages that our proposed method has to offer:
> - Cross-stitch networks calculate a linear combination of activations across a parallel set of layers. Thus, the full model is required to compute the activation for an individual task, and computation grows as a function of the number of tasks. Our work allows pruning down to what is needed for an individual task.
> - PackNet also does pruning, but does so by iteratively pruning and retraining a model for each new task. This raises challenges that do not occur in our work. The amount of the network available to be retrained drops as each new task is added. Furthermore, the order in which tasks are trained plays a significant role in final performance. As seen in their paper, accuracy drops by a couple percentage points for a task if it is added second instead of first, and a couple more if added third instead of second. It is unclear how well this would scale to a large number of tasks, and how much capacity remains in the network for each successive retraining step.
>
> For comparison to other MTL algorithms, here we report the test performance of our method which allows a comparison to two recent works that have evaluated on Visual Decathlon. For context, these methods work by freezing an ImageNet model and adding additional layers for each task to augment intermediate activations. We see that our performance is comparable to these methods despite using less computation and parameters per task. Note, we use the ImageNet pretrained model provided by Rebuffi et al. 2018, so our base architecture is identical to theirs.
>
> Decathlon test set results:
>
>       Airc,   C100, DPed, DTD,   GTSR, Flwr,  Oglt,   SVHN, UCF
> [1] 60.34, 82.12, 92.82, 55.53, 99.42, 81.41, 89.12, 96.55, 51.20
> [2] 64.11, 80.07, 91.29, 56.54, 98.46, 86.05, 89.67, 96.77, 49.38
> [3] 66.04, 81.86, 94.23, 57.82, 99.24, 85.74, 89.25, 96.62, 52.50
> [4] 69.31, 78.81, 91.13, 56.17, 99.14, 85.09, 89.85, 96.41, 51.52
>
> [1] separate networks finetuned for each task (as reported by Rebuffi et al. 2018)
> [2] Rosenfeld et al. 2017
> [3] Rebuffi et al. 2018
> [4] ours

---

### Official Review · AnonReviewer3 · 2019-10-23
**Official Blind Review #3**

**Rating:** 3

**Review:**

This paper proposes a feature partitioning scheme for efficient multi-task neural architecture search.  The proposed scheme automatically determines the network capacity that should be shared across tasks and kept exclusive for each task.

Overall, this is a well written paper.

The problem of neural architecture search is important and beneficial to deep learning community to be able to extract the best out of these methods. Multi-task learning is no exception. However, I am not sure why the problem of NAS is so different for multiple outs than a single output. Given multiple outputs, we need to either look at a weighted combination where weights could be provided by user to reflect the priority over the tasks, or individual outputs in a multi-objective approach such as Pareto optimality manner. I find the approach of this paper a heuristic.

Usually, due to negative transfer learning, too much sharing of the weights is detrimental for the performance of an individual task. I could not see this anywhere in the results. Especially, I missed any baselines where such problems were present, which would then be improved due to using this method. How would the results be if all tasks are given equal weight and NAS is performed with respect to their average output (of course normalised appropriately)?


**Experience Assessment:**

I have read many papers in this area.

**Review Assessment: Checking Correctness Of Derivations And Theory:**

I did not assess the derivations or theory.

**Review Assessment: Checking Correctness Of Experiments:**

I assessed the sensibility of the experiments.

**Review Assessment: Thoroughness In Paper Reading:**

I read the paper at least twice and used my best judgement in assessing the paper.

---

> ### Author Response · Authors · 2019-11-15
> **reply**
>
> Thank you for your comments and taking the time to review.
>
> Deciding on a weighting or prioritization over tasks when doing NAS is orthogonal to this work. In our approach, all tasks are treated with equal importance during optimization. While it would be interesting to look at the trade-offs that occur when weighting tasks differently, that is not the focus of this paper.
>
> Instead, our contributions address parts of the NAS problem that only arise in multi-task settings. Specifically, we search over the different ways features and computation can overlap across tasks. This does not exist in a single-task setting. With a large number of tasks, there are many ways that features could be shared, and this space is quite complicated to search over. The proposed contributions show a principled approach to simplify the space and make the search tractable.
>
> Regarding negative transfer, this is in fact the exact phenomena illustrated in Figure 5. It shows how individual task performance falls as more features are shared across tasks. The y-axis is individual task performance, the x-axis is the amount of parameters allocated to that task, and color indicates how much parameters are shared (lighter = more sharing). We observe that tasks are negatively affected the more they are forced to share with others.

---

### Official Review · AnonReviewer1 · 2019-10-25
**Official Blind Review #1**

**Rating:** 3

**Review:**

This paper proposes a framework for learning multi-task convolutional neural networks. For each layer of the network, the proposed algorithm assigns a subset of the layer's channels to each of the tasks. This is in contrast to existing methods that assign whole layers to tasks. There are two key ideas here: (1) instead of searching in the space of binary assignments of layers to tasks, search in the continuous space of fractions of channels assigned to each layer, subject to some consistency constraints; this allows for using finite differences for gradient estimation which can be fed into a black-box optimization procedure; (2) the use of distillation to estimate the performance of a given assignment, rather than retraining many models. Experimentally, the proposed framework performs relatively well on the Visual Decathlon benchmark.

Overall, I do like the paper's key ideas. However, I am not convinced by the presentation of the paper and by the lack of multi-task learning baselines in the experiments. I will consider changing my score if the following comments are addressed.

- Testing your main premise: "we instead partition out individual feature channels within a layer. This offers a greater degree of control over both the computation required by each task and the sharing that takes place between tasks."
Actually, based on the paper, it is not clear that assigning channels rather than layers brings about additional gains. To prove that hypothesis, you should carry out experiments comparing your method to existing multi-task learning methods that you surveyed in the paper.

- "share all" performance: this baseline seems to do quite well in Table 2, beating es in 5/10 w.r.t. the average accuracy. Why does this happen?

- Too many hacks hidden in the appendix and it is not clear what works why. Have these values been found by cross-validation? The reader should be able to understand how your method works *exactly*. Examples of these magical values:
-- Number of samples for the ES
-- Number of parameter directions
-- "Distillation training is done for a brief 3000 iterations with a batch size of 4 and a learning rate of 1 which is dropped by a factor of 10 at iteration 2000."

Clarification questions:
- I don't understand what "strategy" means here: "Fixed vs Learned Strategies: Is a uniform strategy applied across tasks or is a task-specific solution learned?"

- Section 4.2 needs to be rewritten to give a more structured exposition of the distillation process. Please add pseudocode if needed. The Figure is not very informative.

- MIP: Generally, this is a very important component of your method, yet it is lacking in detail and left to the appendix. Some questions:

-- how long does it take to solve? Which commercial solver do you use? Please provide more details.
-- constraint (8) is for which k? Is it for all k? Please fix. Also for (10), it should be for all i and j as well as k.
-- what is the "non-linear constraint" that (8-9) are linearizing? Please provide the original MIP formulation (without slack or linearization), and explain the final formulation accordingly.
-- MIP: are you adding slack variables to allow for slightly infeasible solutions in case no M can be derived from \tilde{P}?

Minor:
- "cut down average feature dramatically": unclear, rephrase as needed.
- Figure 6 is low-resolution.

**Experience Assessment:**

I do not know much about this area.

**Review Assessment: Checking Correctness Of Derivations And Theory:**

N/A

**Review Assessment: Checking Correctness Of Experiments:**

I carefully checked the experiments.

**Review Assessment: Thoroughness In Paper Reading:**

I read the paper thoroughly.

---

> ### Author Response · Authors · 2019-11-15
> **reply**
>
> Thank you for your detailed comments and feedback.
>
> We provide some comments about comparisons to other methods and offer some additional results in our response to Reviewer 2.
>
> Why does the share all baseline do so well?
> This is actually an interesting finding of this investigation, and part of the reason why we also focus on reducing per-task computation. It turns out if you share everything - even across a large set of tasks - performance will fare incredibly well. This is due to a number of factors:
> - We take a number of measures to ensure that joint optimization over all tasks is as effective as possible (careful tuning of the curriculum, separate momentum terms, etc). All of these factors boost performance whether or not we use feature partitioning.
> - Separate batch normalization statistics are used for each task. This plays a large role in allowing a fully shared network to accommodate multiple image domains and tasks.
> - Ultimately, all tasks are image classification tasks, and while domains differ dramatically across Decathlon, there is likely to be significant overlap in how useful features may be extracted across all of these images.
>
> Too many hacks:
> Apologies for the lack of detail justifying these decisions. We will revise the paper and elaborate further. The majority of decisions are the result of extensive tests and tuning. For example, the distillation training process was tuned to balance performance and training speed. Numerous experiments were done comparing different training procedures at different lengths of time with different batch sizes, and the highest performing amongst these were then validated against a sampling of full training trials (a different set than those used for computing the correlation in Table 1).
>
> Thanks for the comments regarding clarity we will address these. In addition we plan on releasing code to address any issues around reproducibility.
>
> One note, we find empirically that the derivation procedure of M from \tilde{P} is not too important. We tested a few different methods and found no statistically significant difference in performance given different matrices M derived from the same \tilde{P}. That is, even simple heuristics could provide a good enough M that performance was indistinguishable from spending more time to carefully satisfy the constraints perfectly. There is always a bit of noise in the training process and changing a mask M by a fraction of channels one way or the other will not make a noticeable difference in performance (on a related note, it is important to tune the finite differences strategy such that sufficiently different constraint matrices P are compared). This is a discussion that we will make sure to include and elaborate on in the paper.

---

### Decision · Program_Chairs · 2019-12-19

**Decision:**

Reject

**Comment:**

This paper considers how to create efficient architectures for multi-task neural networks. R1 recommends Weak Reject, identifying concerns about the clarity of writing, unsupported claims, and missing or unclear technical details. R2 recommends Weak Accept but calls this a "borderline" case, and has concerns about experiments and comparisons to baselines. R3 also has concerns about experiments and baselines, and feels the approach is somewhat ad hoc. The authors submitted a response that addressed some of these issues, but the authors chose to maintain their decisions. The AC feels the paper has merit but given these slightly negative to borderline reviews, we cannot recommend acceptance at this time. We hope the reviewer comments help the authors to prepare a revision for another venue.